# Detecting and Diagnosing Adversarial Images with Class-Conditional Capsule Reconstructions

**Yao Qin**[*]
UC San Diego
yaq007@eng.ucsd.edu

**Nicholas Frosst**[*]
Google Brain
frosst@google.com

**Sara Sabour**
Google Brain
sasabour@google.com

**Colin Raffel**
Google Brain
craffel@google.com

**Garrison Cottrell**
UC San Diego
gary@eng.ucsd.edu

**Geoffrey Hinton**
Google Brain
geoffhinton@google.com

## ABSTRACT

Adversarial examples raise questions about whether neural network models are sensitive to the same visual features as humans. In this paper, we first *detect* adversarial examples or otherwise corrupted images based on a class-conditional reconstruction of the input. To specifically attack our detection mechanism, we propose the Reconstructive Attack which seeks both to cause a misclassification and a low reconstruction error. This reconstructive attack produces undetected adversarial examples but with much smaller success rate. Among all these attacks, we find that CapsNets always perform better than convolutional networks. Then, we *diagnose* the adversarial examples for CapsNets and find that the success of the reconstructive attack is highly related to the visual similarity between the source and target class. Additionally, the resulting perturbations can cause the input image to appear visually more like the target class and hence become non-adversarial. This suggests that CapsNets use features that are more aligned with human perception and have the potential to address the central issue raised by adversarial examples.

## 1 INTRODUCTION

Adversarial examples (Szegedy et al., 2013) are inputs that are designed by an adversary to cause a machine learning system to make a misclassification. A series of studies on adversarial attacks have shown that it is easy to cause misclassifications using visually imperceptible changes to an image under $\ell_p$-norm based similarity metrics (Goodfellow et al., 2014; Kurakin et al., 2016; Madry et al., 2017; Carlini & Wagner, 2017b; Goodfellow et al., 2018). Since the discovery of adversarial examples, there has been a constant "arms race" between better attacks and better defenses. Many new defenses have been proposed (Song et al., 2017; Gong et al., 2017; Grosse et al., 2017; Metzen et al., 2017), only to be broken shortly thereafter (Carlini & Wagner, 2017a; Athalye et al., 2018). Hinton et al. (2018) showed that capsule models are more robust to simple adversarial attacks than CNNs but Michels et al. (2019) showed that this is not the case for all attacks.

The cycle of attacks and defenses motivates us to rethink both how we can improve the general robustness of neural networks as well as the high-level motivation for this pursuit. One potential path forward is to detect adversarial inputs, instead of attempting to accurately classify them (Schott et al., 2018; Roth et al., 2019). Recent work (Jetley et al., 2018; Gilmer et al., 2018b) argue that adversarial examples can exist within the data distribution, which implies that detecting adversarial examples based on an estimate of the data distribution alone might be insufficient. Instead, in this paper we develop methods for ***detecting*** adversarial examples by making use of *class-conditional reconstruction networks*. These sub-networks, first proposed by Sabour et al. (2017) as part of a Capsule Network (CapsNet), allow a model to produce a reconstruction of its input based on the identity and instantiation parameters of the winning capsule. Interestingly, we find that reconstructing

---

[*]Equal Contributions.

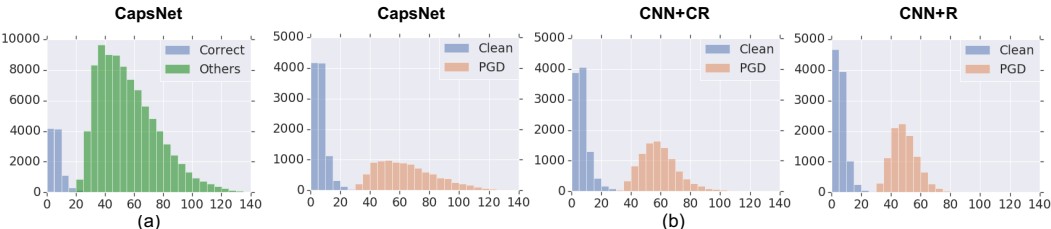

Figure 1: (a) The histogram of $\ell_2$ distances between the input and the reconstruction using the correct capsule or other capsules in CapsNet on the real MNIST images. Notice the stark difference between the distributions of reconstructions of the capsule corresponding to the correct class and other capsules. (b) The histograms of $\ell_2$ distances between the reconstruction and the input for real and adversarial images for the three models explored in this paper on the MNIST dataset. We use PGD (Madry et al., 2017) with the $\ell_\infty$ bound $\epsilon = 0.3$ to create the attacks.

an input from the capsule corresponding to the correct class results in a much lower reconstruction error than reconstructing the input from capsules corresponding to incorrect classes, as shown in Figure 1(a). Motivated by this, we propose using the reconstruction sub-network in a CapsNet as an attack-independent detection mechanism. Specifically, we reconstruct a given input from the pose parameters of the winning capsule and then detect adversarial examples by comparing the difference between the reconstruction distributions for natural and adversarial (or otherwise corrupted) images.

We extend this detection mechanism to standard convolutional neural networks (CNNs) and show its effectiveness against black box and white box attacks on three image datasets; MNIST, Fashion-MNIST and SVHN. We show that capsule models achieve the strongest attack detection rates and accuracy on these attacks. We then test our method against a stronger attack, the Reconstructive Attack, specifically designed to attack our detection mechanism by generating adversarial examples with a small reconstruction error. With this attack we are able to create undetected adversarial examples, but we show that this attack is less successful in fooling the classifier than a non-reconstructive attack.

Among all these attacks, we find CapsNets perform the best in detecting adversarial examples. To explain the success of CapsNets over CNNs, we further ***diagnose*** the adversarial examples for CapsNets and find that 1) the success of the targeted reconstructive attack is highly dependent on the visual similarity between the source image and the target class. 2) many of the resultant attacks resemble members of the target class and so cease to be "adversarial" – i.e., they may also be misclassified by humans. These findings suggest that CapsNets with class conditional reconstructions have the potential to address the real issue with adversarial examples – networks should make predictions based on the same properties of the image that people use rather than using features that can be manipulated by an imperceptible adversarial attack.

In summary, our main contributions are:

- We propose a class-conditional capsule reconstruction based detection method to detect standard white-box/black-box adversarial examples on three datasets. This detection mechanism is attack-agnostic and is successfully extended to standard convolutional neural networks.
- We test our detection mechanism on the corrupted MNIST dataset and show that it can work as a general out-of-distribution detector.
- A stronger reconstructive attack is specifically designed to attack our detection mechanism but becomes less successful in fooling the classifier.
- We perform extensive qualitative studies to explain the superior performance of CapsNets in detecting adversarial examples compared to CNNs. The results suggest that the features captured by CapsNets are more aligned with human perception.

## 2 RELATED WORK

Adversarial examples were first introduced in (Biggio et al., 2013; Szegedy et al., 2013), where a given image was modified by following the gradient of a classifier's output with respect to the

image's pixels. Goodfellow et al. (2014) then developed the more efficient Fast Gradient Sign method (FGSM), which can change the label of the input image $X$ with a similarly imperceptible perturbation that is constructed by taking an $\epsilon$ step in the direction of the gradient. Later, the Basic Iterative Method (BIM) (Kurakin et al., 2016) and Projected Gradient Descent (Madry et al., 2017) can generate stronger attacks improved on FGSM by taking multiple steps in the direction of the gradient. In addition, Carlini & Wagner (2017b) proposed another iterative optimization-based method to construct strong adversarial examples with small perturbations.

An early approach to reducing vulnerability to adversarial examples was proposed by (Goodfellow et al., 2014), where a network was trained on both clean images and adversarially perturbed ones. Since then, there has been a constant "arms race" between better attacks and better defenses; Kurakin et al. (2018) provide an overview of this field. However, many defenses against adversarial examples have been demonstrated to be an effect of "obfuscated gradients" and can be further circumvented under the white-box setting (Athalye et al., 2018).

Another line of work attempts to circumvent adversarial examples by detecting them with a separately-trained classifier (Gong et al., 2017; Grosse et al., 2017; Metzen et al., 2017) or using statistical properties (Hendrycks & Gimpel, 2016; Li & Li, 2017; Feinman et al., 2017; Grosse et al., 2017). However, many of these approaches were subsequently shown to be flawed (Carlini & Wagner, 2017a; Athalye et al., 2018). The most recent work in detecting adversarial examples (Roth et al., 2019) that has a 99% true positive rate on CIFAR-10 dataset (Krizhevsky, 2009) has also been fully bypassed by later work (Hosseini et al., 2019) which decreased the true positive rate to less than 2%.

Similar to our work, Schott et al. (2018) also investigated the effectiveness of a class-conditional generative model as a defense mechanism for MNIST digits. However, we differ in some important ways. Their model is in some ways the opposite of ours - they first attempt to generate the input, and then make a classification on the resulting generated images, whereas our method attempts to first classify the input, making use of an otherwise unchanged capsule classification model, and then generates the input from a high level representation. As such, our method does not increase the computational overhead of classifying the input, compared to the approach of Schott et al. (2018). In addition, the work of Schott et al. (2018) is only applied to MNIST, so our results on the more complex datasets represent an improvement.

## 3    PRELIMINARIES

**Adversarial Examples**    Given a clean test image $x$, its corresponding label $y$, and a classifier $f(\cdot)$ which predicts a class label given an input, we refer to $x' = x + \delta$ as an adversarial example if it is able to fool the classifier into making a wrong prediction $f(x') \neq f(x) = y$. The small adversarial perturbation $\delta$ (where "small" is measured under some norm) causes the adversarial example $x'$ to appear visually similar to the clean image $x$ but to be classified differently. In the unrestricted case where we only require that $f(x') \neq y$, we refer to $x'$ as an "untargeted adversarial example". A more powerful attack is to generate a "targeted adversarial example": instead of simply fooling the classifier to make a wrong prediction, we force the classifier to predict some targeted label $f(x') = t \neq y$. In this paper, the target label $t$ is selected uniformly at random as any label which is not the ground-truth correct label. As is standard practice in the literature, in this paper we test our detection mechanism on three $\ell_\infty$ norm based attacks (fast gradient sign method (FGSM) (Goodfellow et al., 2014), the basic iterative method (BIM) (Kurakin et al., 2016), projected gradient descent (PGD) (Madry et al., 2017)) and one $\ell_2$ norm based attack (Carlini-Wagner (CW) (Carlini & Wagner, 2017b)).

**Capsule Networks**    Capsule Networks (CapsNets) are an alternative architecture for neural networks (Sabour et al., 2017; Hinton et al., 2018). In this work we make use of the CapsNet architecture detailed by (Sabour et al., 2017). Unlike a standard neural network which is made up of layers of scalar-valued units, CapsNets are made up of layers of capsules, that output a vector or matrix. Intuitively, just as one can think of the activation of a unit in a normal neural network as the presence of a feature in the input, the activation of a capsule can be thought of as both the presence of a feature and the pose parameters that represent attributes of that feature. A top-level capsule in a classification network therefore outputs both a classification and pose parameters that represent the instance of that class in the input. This high level representation allows us to train a reconstruction network.

**Threat Model**    In this paper, we test our detection mechanism against both white-box and black-box attacks. For white-box attacks, the adversary has full access to the model as well as its parameters. In particular, the adversary is allowed to compute the gradient through the model to generate adversarial examples. To perform black-box attacks, the adversary is allowed to know the network architecture but not its parameters. Therefore, we retrain a substitute model that has the same architecture as the target model and generate adversarial examples by attacking the substitute model. Then we transfer these attacks to the target model. For $\ell_\infty$ based attacks, we always control the $\ell_\infty$ norm of the adversarial perturbation to be within a relatively small bound $\epsilon_\infty$, specific to each dataset.

## 4    DETECTING ADVERSARIAL IMAGES BY RECONSTRUCTION

To detect adversarial images, we make use of the reconstruction network proposed in (Sabour et al., 2017), which takes pose parameters $v$ as input and outputs the reconstructed image $r(v)$. The reconstruction network is simply a fully connected neural network with two ReLU hidden layers with 512 and 1024 units respectively, with a sigmoid output with the same dimensionality as the dataset. The reconstruction network is trained to minimize the $\ell_2$ distance between the input image and the reconstructed image. This same network architecture is used for all the models and datasets we explore. The only difference is what is given to the reconstruction network as input.

### 4.1    MODELS

**CapsNet**    The reconstruction network of the CapsNet is *class-conditional*: It takes in the pose parameters of all the class capsules and masks all values to 0 except for the pose parameters of the predicted class. We use this reconstruction network for detecting adversarial attacks by measuring the Euclidean distance between the input and a class conditional reconstruction. Specifically, for any given input $x$, the CapsNet outputs a prediction $f(x)$ as well as the pose parameters $v$ for all classes. The reconstruction network takes in the pose parameters and then selects the pose parameter corresponding to the predicted class, denoted as $v_{f(x)}$, to generate a reconstruction $r(v_{f(x)})$. Then we compute the $\ell_2$ reconstruction distance $d(x) = \|r(v_{f(x)}), x\|_2$ between the reconstructed image and the input image, and compare it with a pre-defined detection threshold $\theta$ (described below in Section 4.2). If the reconstruction distance $d(x)$ is higher than the detection threshold $\theta$, we flag the input as an adversarial example. Figure 1 (b) shows an example of histograms of reconstruction distances for natural images and typical adversarial examples.

**CNN+CR**    Although our strategy is inspired by the reconstruction networks used in CapsNets, the strategy can be extended to standard convolutional neural networks (CNNs). We create a similar architecture, CNN with conditional reconstruction (CNN+CR), by dividing the penultimate hidden layer of a CNN into groups corresponding to each class. The sum of each neuron group serves as the logit for that particular class and the group itself serves the same purpose as the pose parameters in the CapsNet. We use the same masking mechanism as Sabour et al. (2017) to select the pose parameter corresponding to the predicted label $v_{f(x)}$ and generate the reconstruction based on the selected pose parameters. In this way we extend the class-conditional reconstruction network to standard CNNs.

**CNN+R**    We can also create a more naïve implementation of our strategy by simply computing the reconstruction from the activations in the entire penultimate layer without any masking mechanism. We call this model the "CNN+R" model. In this way we are able to study the effect of conditioning on the predicted class.

### 4.2    DETECTION THRESHOLD

We find the threshold $\theta$ for detecting adversarial inputs by measuring the reconstruction error between a validation input image and its reconstruction. If the distance between the input and the reconstruction is above the chosen threshold $\theta$, we classify the data as adversarial. Choosing the detection threshold $\theta$ involves a trade-off between false positive and false negative detection rates. The optimal threshold depends on the probability of the system being attacked. Such a trade-off is discussed by Gilmer et al. (2018a). In our experiments we don't tune this parameter and simply set it as the 95th percentile of validation distances. This means our false positive rate on real validation data is $5\%$.

Table 1: **Success Rate / Undetected Rate** of white-box targeted and untargeted attacks on the MNIST dataset. In the table, $S_t/R_t$ is shown for targeted attacks and $S_u/R_u$ is presented for untargeted attacks. A smaller success rate and undetected rate means a stronger defense model. Full results for FashionMNIST and SVHN can be seen in Table 5 in the Appendix.

| Networks | Targeted (%) | | | | Untargeted (%) | | | |
|---|---|---|---|---|---|---|---|---|
| | FGSM | BIM | PGD | CW | FGSM | BIM | PGD | CW |
| CapsNet | **3/0** | **82/0** | **86/0** | **99/2** | **11/0** | **99/0** | **99/0** | **100/19** |
| CNN+CR | 16/0 | 93/0 | 95/0 | 89/8 | 85/0 | 100/0 | 100/0 | 100/28 |
| CNN+R | 37/0 | 100/0 | 100/0 | 100/47 | 64/0 | 100/0 | 100/0 | 100/63 |

### 4.3 EVALUATION METRICS

We use **Success Rate** to measure the success of attacks. For targeted attacks, the success rate $S_t$ is defined as the proportion of inputs which are classified as the target class, $S_t = \frac{1}{N} \sum_i^N (f(x'_i) = t_i)$, while the success rate for untargeted attacks is defined as the proportion of inputs which are misclassified, $S_u = \frac{1}{N} \sum_i^N (f(x'_i) \neq y_i)$. Previous work (Carlini & Wagner, 2017a; Hosseini et al., 2019) used the **True Positive Rate** to measure the proportion of adversarial examples that are detected, which alone is insufficient to measure the ability of different detection mechanism because the unsuccessful adversarial examples do not have to be detected. Therefore, in this paper, we propose to use the **Undetected Rate**: the proportion of attacks that are successful and undetected to evaluate the detection mechanism. For targeted attacks, the undetected rate is defined as $R_t = \frac{1}{N} \sum_i^N (f(x'_i) = t_i) \cap (d(x'_i) \leq \theta)$, where $d(\cdot)$ computes the reconstruction distance of the input and $\theta$ denotes the detection threshold introduced in Section 4.2. Similarly, the undetected rate for untargeted attacks $R_u$ can be defined as $R_u = \frac{1}{N} \sum_i^N (f(x'_i) \neq y_i) \cap (d(x'_i) \leq \theta)$. The smaller the undetected rate $R_t$ or $R_u$ is, the stronger the model is in detecting adversarial examples. The undetected rate can also be used to evaluate the attacks (higher is better). We also plot the **Undetected Rate vs. False Positive Rate** curve to compare the detection performance between different models, where **False Positive Rate** is defined as the proportion of clean examples that are misclassified as the adversarial example by the detection method.

### 4.4 TEST MODELS AND DATASETS

In all experiments, all three models (CapsNet, CNN+R, and CNN+CR) have the same number of parameters and were trained with Adam (Kingma & Ba, 2014) for the same number of epochs. In general, all models achieved similar test accuracy. We did not do an exhaustive hyperparameter search on these models, instead we chose hyperparameters that allowed each model to perform roughly equivalently on the test sets. We run experiments on three datasets: MNIST (LeCun et al., 1998), FashionMNIST (Xiao et al., 2017), and SVHN (Netzer et al., 2011). The test error rate for each model on these three datasets, as well as details of the model architectures, can be seen in Section A and Section B in the Appendix.

## 5 EXPERIMENTS

We first demonstrate how reconstruction networks can detect standard white and black-box attacks in addition to naturally corrupted images. Then, we introduce the "reconstructive attack", which is specifically designed to circumvent our defense and show that it is a more powerful attack in this setting. Based on this finding, we qualitatively study the kind of misclassifications caused by the reconstructive attack and argue that they suggest that CapsNets learn features that are better aligned with human perception.

### 5.1 STANDARD ATTACKS

**White Box**  We present the success and undetected rates for several targeted and untargeted attacks on MNIST (Table 1), FashionMNIST, and SVHN (Table 5 presented in the Appendix). Our method

Table 2: **Error Rate/Undetected Rate** on the Corrupted MNIST dataset. A smaller error rate and undetected rate means a better defense model.

| Corruption | Clean | Gaussian Noise | Gaussian Blur | Line | Dotted Line | Elastic Transform |
|---|---|---|---|---|---|---|
| CapsNet | 0.6/0.2 | 12.1/0.0 | 10.3/4.1 | 19.6/0.1 | 4.3/0.0 | 11.3/0.8 |
| CNN+CR | 0.7/0.3 | 9.8/0.0 | 6.7/4.2 | 17.6/0.1 | 4.2/0.0 | 11.1/1.1 |
| CNN+R | 0.6/0.4 | 6.7/0.0 | 8.9/6.4 | 18.9/0.1 | 3.1/0.0 | 12.2/2.1 |
| **Corruption** | **Saturate** | **JPEG** | **Quantize** | **Sheer** | **Spatter** | **Rotate** |
| CapsNet | 3.5/0.0 | 0.8/0.4 | 0.7/0.1 | 1.6/0.4 | 1.9/0.2 | 6.5/2.2 |
| CNN+CR | 1.5/0.0 | 0.8/0.5 | 0.9/0.1 | 2.1/0.4 | 1.8/0.4 | 6.1/1.6 |
| CNN+R | 1.2/0.0 | 0.7/0.5 | 0.7/0.2 | 2.2/0.7 | 1.8/0.4 | 6.5/3.4 |
| **Corruption** | **Contrast** | **Inverse** | **Canny Edge** | **Fog** | **Frost** | **Zigzag** |
| CapsNet | 92.0/0.0 | 91.0/0.0 | 21.5/0.0 | 83.7/0.0 | 70.6/0.0 | 16.9/0.0 |
| CNN+CR | 72.0/32.6 | 78.1/0.0 | 34.6/0.0 | 66.0/0.5 | 37.6/0.0 | 18.4/0.0 |
| CNN+R | 73.4/49.4 | 88.1/0.0 | 23.4/0.0 | 65.6/0.1 | 36.2/0.0 | 17.5/0.0 |

is able to accurately detect many attacks with very low undetected rates. Capsule models almost always have the lowest undetected rates out of our three models. It is worth noting that this method performs best with the simplest dataset, MNIST, and that the highest undetected rates are found with the Carlini-Wagner attack on the SVHN dataset. This illustrates both the strength of this attack and a shortcoming of our defense, namely that our detection mechanism relies on $\ell_2$ image distance as a proxy for visual similarity, and in the case of higher dimensional color datasets such as SVHN, this proxy is less meaningful.

**Black Box**  We also tested our detection mechanism results on black box attacks. Given the low undetected rates in the white-box settings, it is not surprising that our detection method is able to detect black box attacks as well. In fact, on the MNIST dataset the capsule model is able to detect all targeted and untargeted PGD attacks. Both the CNN-R and the CNN-CR models are able to detect the black box attacks as well, but with a relatively higher undetected rate. A table of these results can be seen in Table 7 in the Appendix.

## 5.2 CORRUPTION ATTACKS

Recent work has argued that improving the robustness of neural networks to $\ell_p$ norm bounded adversarial attacks should not come at the expense of increasing error rates under distributional shifts that do not affect human classification rates and are likely to be encountered in the "real-world" (Gilmer et al., 2018a). For example, if an image is corrupted due to adverse weather, lighting, or occlusion, we might hope that our model can continue to provide reliable predictions or detect the distributional shift. We can test our detection method on its ability to detect these distributional shifts by making use of the Corrupted MNIST dataset (Mu & Gilmer, 2019). This data set contains many visual transformations of MNIST that do not seem to affect human performance, but nevertheless are strongly misclassified by state-of-the-art MNIST models. Our three models can almost always detect these distributional shifts (in all corruptions CapsNets have either a small undetected rate or an undetected rate of 0). The error rate (the proportion of misclassified input) and undetected rate of three test models on the Corrupted MNIST dataset is shown in Table 2. Please refer to Figure 7 and Figure 8 in the Appendix for visualization of Corrupted MNIST.

## 5.3 RECONSTRUCTIVE ATTACKS

Thus far we have only evaluated previously-defined attacks. Following the suggestion in (Carlini & Wagner, 2017a) that detection methods need to show effectiveness towards defense-aware attacks, we introduce an attack specifically designed to take into account our defense mechanism. In order to construct adversarial examples that cannot be detected by the network, we propose a two-stage optimization method to generate a "reconstructive attack".

Table 3: Success rate and the **worst case** undetected rate of white-box targeted and untargeted reconstructive attacks. $S_t/R_t$ is shown for targeted attacks and $S_u/R_u$ is presented for untargeted attacks. The worst case undetected rate is reported via tuning the hyperparameter $\beta$ in Eqn 1 and Eqn 2. The best defense models are shown in **bold** (smaller success rate and undetected rate is better). All the numbers are shown in %. A full table with more attacks can be seen in Table 6 in Appendix.

|  | MNIST | | FASHION | | SVHN | |
|---|---|---|---|---|---|---|
|  | Targeted R-PGD | Untargeted R-PGD | Targeted R-PGD | Untargeted R-PGD | Targeted R-PGD | Untargeted R-PGD |
| **CapsNet** | **50.7/33.7** | **88.1/37.9** | **53.7/29.8** | **84.9/75.5** | **82.0/79.2** | **98.9/97.5** |
| **CNN+CR** | 98.6/68.1 | 99.4/87.7 | 89.8/84.4 | 91.5/86.0 | 99.0/97.9 | 99.9/99.5 |
| **CNN+R** | 95.5/71.2 | 95.1/70.5 | 94.6/88.4 | 98.9/90.0 | 99.5/99.3 | 100.0/99.9 |

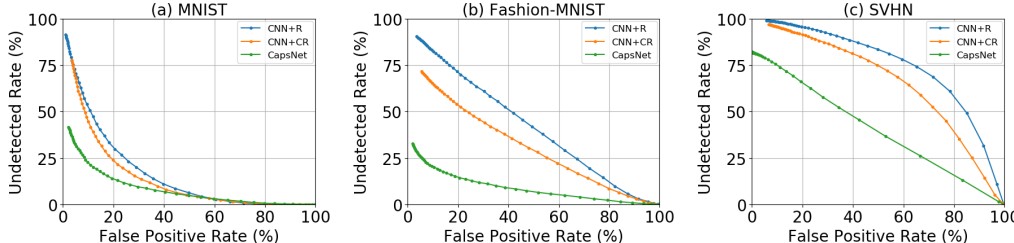

Figure 2: The undetected rate of the white-box targeted defense-aware R-PGD attack versus the False Positive Rate on the MNIST, Fashion-MNIST and SVHN datasets.

**Untargeted Reconstructive Attacks**   To construct untargeted reconstructive attacks, we first update the perturbation based on the gradient of the cross-entropy loss function following a standard FGSM attack (Goodfellow et al., 2014), that is:

$$\delta \leftarrow \text{clip}_\epsilon(\delta + c \cdot \beta \cdot \text{sign}(\nabla_\delta \ell_{net}(f(x + \delta), y))), \tag{1}$$

where $\ell_{net}(f(\cdot), y)$ is the cross-entropy loss function, $\epsilon$ is the $\ell_\infty$ bound for our attacks, $c$ is a hyperparameter controlling the step size in each iteration and $\beta$ is a hyperparameter which balances the importance of the cross-entropy loss and the reconstruction loss (explained further below). In the second stage, we focus on constraining the reconstructed image from the newly predicted label to have a small reconstruction distance by updating $\delta$ according to

$$\delta \leftarrow \text{clip}_\epsilon(\delta - c \cdot (1 - \beta) \cdot \text{sign}(\nabla_\delta(\|r(v_{f(x+\delta)}) - (x + \delta)\|_2))), \tag{2}$$

where $r(v_{f(x+\delta)})$ is the class-conditional reconstruction based on the predicted label $f(x + \delta)$ in a CapsNet or CNN+CR network. The $\delta$ used here is the optimized $\delta$ from the first stage. $\|r(v_{f(x+\delta)}) - (x + \delta)\|_2$ is the $\ell_2$ reconstruction distance between the reconstructed image and the input image. Since the CNN+R network does not use the class conditional reconstruction, we simply use the reconstructed image without the masking mechanism. According to Eqn 1 and Eqn 2, we can see that $\beta$ balances the importance between the success rate of attacks and the reconstruction distance. This hyperparameter was tuned for each model and each dataset in order to create the strongest attacks. The success rate and undetected rate change as this parameter, which is shown in Figure 9 in Appendix.

**Targeted Reconstructive Attacks**   We perform a similar two-stage optimization to construct targeted reconstructive attacks, by defining a target label and attempting to maximize the classification probability of this label, and minimize the reconstruction error from corresponding capsule. Because the targeted label is given, another way to construct targeted reconstructive attacks is to combine these two stages into one stage via minimizing the loss function $\ell = \beta \cdot \ell_{net}(f(x + \delta), y) + (1 - \beta) \cdot \|r(v_{f(x+\delta)}) - (x + \delta)\|_2$. We implemented both of these targeted reconstructive attacks and found that the two-stage version is a stronger attack. Therefore, all the Reconstructive Attack experiments performed in this paper are based on two-stage optimization.

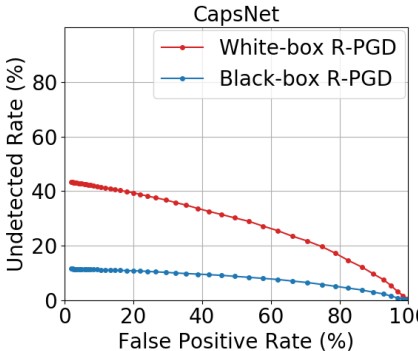 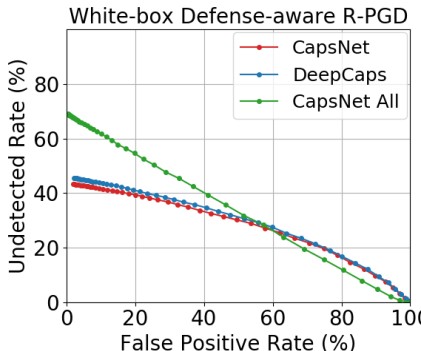

Figure 3: The defense-aware R-PGD attack is tested on the CIFAR-10 dataset with $\epsilon_\infty = 8/255$. **Left**: The undetected rate of white-box/black-box defense-aware R-PGD versus the Fasle Positive Rate for the clean examples. The test model is our CapsNet. **Right**: The undetected rate of white-box defense-aware R-PGD versus the Fasle Positive Rate for the clean examples. The test model is our CapsNet using class-conditional reconstruction, "CapsNet All" using all capsule information, and the DeepCaps (Rajasegaran et al., 2019) using class-independent capsule information.

We build our reconstructive attack based on the standard PGD attack, denoted as R-PGD, and test the performance of our detection models against this reconstructive attack in a white-box setting (white-box Reconstructive FGSM and BIM are reported in Table 6 in the Appendix). Comparing Table 1 and Table 3, we can see that the Reconstructive Attack is significantly less successful at changing the models prediction (lower success rates than the standard attack). However, this attack is more successful at fooling our detection method. For all attacks and datasets the capsule model has the lowest attack success rate and the lowest undetected rate. We report results for black-box R-PGD attacks in Table 7 in the Appendix, which suggest similar conclusions.

In addition, we report the undetected rate of the white-box targeted defense-aware R-PGD attack versus the False Positive Rate on the MNIST, Fashion-MNIST and SVHN datasets in Figure 2. We can clearly see that the undetected rate of the defense-aware attack against CapsNet is significantly smaller than the CNN-based networks, which suggests that CapsNets are more robust against adversarial attacks. Furthermore, CNN with class-conditional reconstruction (CNN+CR) has smaller undetected rate at the same False Positive Rate compared to the CNN without class-conditional reconstruction (CNN+R), which suggests the class-conditional information is helpful in our models to improve the robustness against adversarial attacks.

## 5.4 CIFAR-10 DATASET

In order to show that our method based on CapsNet is capable to scale up to more complex datasets, we test our detection method with a deeper reconstruction network on CIFAR-10 (Krizhevsky, 2009). The classification accuracy on the clean test dataset is $92.2\%$. In addition, we display the undetected rate of the white-box/black-box defense-aware R-PGD attack against CapsNets versus the False Positive Rate in Figure 3 (Left), where we can see a significant drop of the undetected rate of black-box R-PGD compared to the white-box setting. This indicates the CapsNets greatly reduce the attack transferability and the threat of black-box attacks.

**Class-conditional Information** To investigate the effectiveness of the class-conditional information in the reconstruction network, we compare our CapsNet based on (Sabour et al., 2017) with the other two variants of CapsNets: "CapsNet All" and "DeepCaps" (Rajasegaran et al., 2019). In "CapsNet All", we remove the masking mechanism in the CapsNet and use all the capsules to do the reconstruction. In "DeepCaps", we extract the winning-capsule information as a single vector and used it as the input for the reconstruction network instead of using a masking mechanism to mask out the losing capsules information. In this way, the class information in DeepCaps is more explicitly fed into the reconstruction network. As shown in Figure 3 (right), our CapsNet has the best detection performance (the lowest undetected rate at the same False Positive Rate) compared to the other two

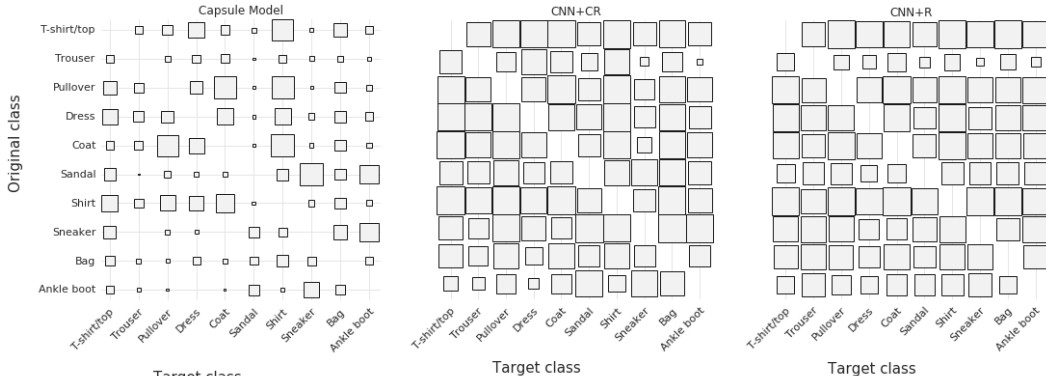

Figure 4: This diagram visualizes the adversarial success rates for each source/target pair for targeted R-PGD attacks on Fashion-MNIST with $\epsilon_\infty = 25/255$. The size of the box at position x, y represents the success rate of adversarially perturbing inputs of class x to be classified as class y. We can see that there is significantly higher variance for the CapsNet model than for the two CNN models.

Figure 5: These are randomly sampled (not cherry picked) successful and undetected adversarial attacks created by R-PGD with a target class of 0 for each model on the SVHN dataset($\epsilon_\infty = 25/255$). We can see that for the capsule model, many of the attacks are not "adversarial" as they resemble members of the target class.

Capsule models. "DeepCaps" performs slightly worse than our "CapsNet" and "CapsNet All" has the worst detection performance. Therefore, we conclude that the class-conditional information used in the reconstruction network increases the model's robustness to adversarial attack. This also holds true to CNN-based networks because CNN+CR has a better detection performance than CNN+R, shown in Figure 2.

## 6 VISUAL COHERENCE OF THE RECONSTRUCTIVE ATTACK

The great success of CapsNet over CNN-based models motivates us to further diagnose the generated adversarial examples for CapsNets. If our true aim in adversarial robustness research is to create models that make predictions based on reasonable and human-observable features, then we would prefer models that are more likely to misclassify a "shirt" as a "t-shirt" (in the case of FashionMNIST) than to misclassify a "bag" as a "sweater". For a model to behave ideally, the success of an adversarial perturbation would be related to the visual similarity between the source and the target class. By visualizing a matrix of adversarial success rates between each pair of classes (shown in Figure 4), we can see that for the capsule model there is a great variance between the source and target class pairs and that the success rate of attacks is highly related to the visual similarity of the classes. However, this is not the case for either of the other two CNN-based models.

Thus far we have treated all attacks as equal. However, a key component of an adversarial example is that it is visually similar to the source image, and that it does not resemble the adversarial target class. The adversarial research community makes use of a small epsilon bound as a mechanism for ensuring that the resultant adversarial attacks are visually unchanged from the source image. For standard attacks against CNN-based models this heuristic is sufficient, because taking gradient steps in the image space in order to have a network misclassify an image normally results in something

visually similar to the source image. But this is not the case for adversarial attacks against CapsNets. As shown in Figure 5, when we use R-PGD to attack the CapsNet, many of the resultant attacks resemble members of the target class. In this way, they stop being "adversarial". As such, an attack detection method which does not detect them as adversarial is arguably behaving correctly. This puts the previously undetected rates presented earlier in a new light, and illustrates a difficulty in the evaluation of adversarial attacks and defenses. In addition, it should be noted that this phenomenon rarely occurs in a standard convolutional neural network, which suggests that the features captured by CapsNet are more aligned with human perception.

## 7 DISCUSSION

Our detection mechanism relies on a similarity metric (i.e. a measure of reconstruction error) between the reconstruction and the input. This metric is required both during training in order to train the reconstruction network and during test time in order to flag adversarial examples. In the four datasets we have evaluated, the distance between examples roughly correlates with semantic similarity. However, this may not be the case for images in more complex datasets such as the SUN dataset (Xiao et al., 2010) and ImageNet (Deng et al., 2009), in which two images may be similar in terms of semantic content but nevertheless have significant $\ell_2$ distance. A better similarity metric (Theis et al., 2015; Zhang et al., 2018) can be further explored to extend our methods to more complex problems. Furthermore our reconstruction network is trained on a hidden representation of one class but is trained to reconstruct the entire input. In datasets without distractors or backgrounds, this is not a problem. But in the case of ImageNet, in which the object responsible for the classification is not the only object in the image, attempting to reconstruct the entire input from a class encoding seems misguided.

## 8 CONCLUSION

We have presented a class-conditional reconstruction-based detection method that does not rely on a specific predefined adversarial attack. We have shown that by reconstructing the input from the internal class-conditional representation, our system is able to accurately detect black-box and white-box FGSM, BIM, PGD, and CW attacks. We then proposed a new attack to beat our defense - the Reconstructive Attack - in which the adversary optimizes not only the classification loss but also minimizes the reconstruction loss. We showed that this attack was able to fool our detection mechanism but with a much smaller success rate than a standard attack.

Compared to CNN-based models, we showed that the CapsNet was able to detect adversarial examples with greater accuracy on all the datasets we explored. To further explain the success of CapsNet, we qualitatively showed that the success of the reconstructive attack was highly related to the visual similarity between the target class and the source class for the CapsNet. In addition, we showed that images generated by this reconstructive attack to attack the CapsNet are not typically adversarial, i.e. many of the resultant attacks resemble members of the target class even with a small $\ell_\infty$ norm bound. These are not the case for the CNN-based models. The extensive qualitative studies indicate that the capsule model relies on visual features similar to those used by humans. We believe this is a step towards solving the true problem posed by adversarial examples.

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

APPENDIX

## A NETWORK ARCHITECTURES

Figure 6 shows the architecute of the capsule network, the CNN reconstruction model and the CNN conditional reconstruction model used for experiments on MNIST, FashionMNIST and SVHN dataset. MNIST and Fashion MNIST have exactly the same architectures while we use larger models for SVHN. Note that the only difference between the CNN reconstruction (CNN+R) and the CNN conditional reconstruction (CNN+CR) is the masking procedure on the input to the reconstruction network based on the predicted class. All three models have the same number of parameters.

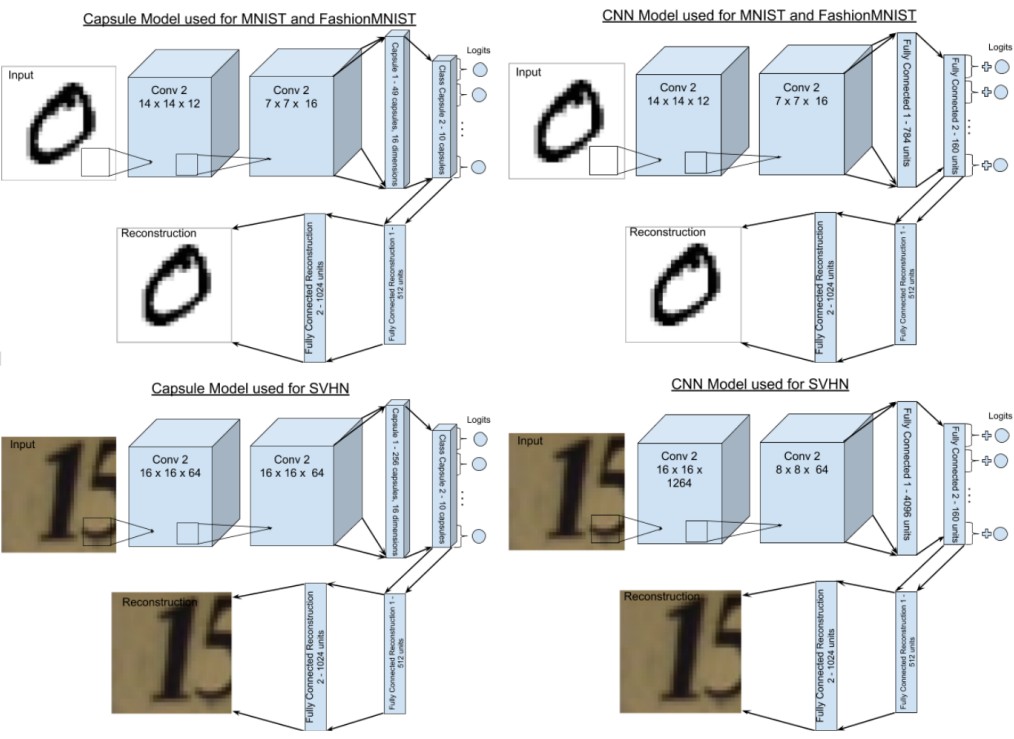

Figure 6: The architecture for the CapsNet, CNN+R and CNN+CR model used for our experiments on MNIST (LeCun et al., 1998), FashionMNIST (Xiao et al., 2017), and SVHN (Netzer et al., 2011).

## B TEST MODELS

The error rate of each test model used in the paper are presented in Table 4. We ensure that they have similar performance.

Table 4: Error rate of each model when the input are clean test images in each dataset.

| Dataset | CapsNet | CNN+CR | CNN+R |
|---|---|---|---|
| MNIST | 0.6% | 0.7% | 0.6% |
| FashionMNIST | 9.6% | 9.5% | 9.3% |
| SVHN | 10.7% | 9.3% | 9.5% |

## C    IMPLEMENTATION DETAILS

For all the $\ell_\infty$ based adversarial examples, the $\ell_\infty$ norm of the perturbations is bound by $\epsilon$, which is set to 0.3, 0.1, 0.1 for MNIST, Fashion MNIST and SVHN dataset respectively following previous work (Madry et al., 2017; Song et al., 2017). In FGSM based attacks, the step size $c$ is 0.05. In BIM-based (Kurakin et al., 2016) and PGD-based (Madry et al., 2017) attacks, the step size $c$ is 0.01 for all the datasets and the number of iterations are 1000, 500 and 200 for MNIST, Fashion MNIST and SVHN dataset respectively. We choose a sufficiently large number of iterations to ensure the attacks has converged.

We use the publicly released code from the authors of (Carlini & Wagner, 2017b) to perform the CW attack for our models. The number of iterations are set to 1000 for all three datasets.

## D    WHITE BOX STANDARD ATTACKS

The results of four white box standard attacks on the three datasets are shown in Table 5.

Table 5: Success rate and undetected rate of white-box targeted and untargeted attacks. In the table, $S_t/R_t$ is shown for targeted attacks and $S_u/R_u$ is presented for untargeted attacks.

| Networks | Targeted (%) | | | | Untargeted (%) | | | |
|---|---|---|---|---|---|---|---|---|
| | FGSM | BIM | PGD | CW | FGSM | BIM | PGD | CW |
| **MNIST Dataset** | | | | | | | | |
| CapsNet | 3/0 | 82/0 | 86/0 | 99/2 | 11/0 | 99/0 | 99/0 | 100/19 |
| CNN+CR | 16/0 | 93/0 | 95/0 | 89/8 | 85/0 | 100/0 | 100/0 | 100/28 |
| CNN+R | 37/0 | 100/0 | 100/0 | 100/47 | 64/0 | 100/0 | 100/0 | 100/63 |
| **FASHION MNIST Dataset** | | | | | | | | |
| CapsNet | 7/5 | 54/9 | 55/10 | 100/26 | 35/29 | 86/50 | 87/51 | 100/68 |
| CNN+CR | 19/13 | 89/28 | 89/28 | 87/37 | 74/33 | 100/25 | 100/24 | 100/72 |
| CNN+R | 23/16 | 98/19 | 98/19 | 99/81 | 62/48 | 100/35 | 100/34 | 100/87 |
| **SVHN Dataset** | | | | | | | | |
| CapsNet | 22/20 | 83/45 | 84/46 | 100/90 | 74/67 | 99/70 | 99/68 | 100/94 |
| CNN+CR | 24/23 | 99/90 | 99/90 | 99/93 | 87/82 | 100/90 | 100/89 | 100/90 |
| CNN+R | 26/24 | 100/86 | 100/86 | 100/94 | 88/82 | 100/92 | 100/92 | 100/95 |

# E  VISUALIZATION OF CORRUPTED MNIST DATASET

Visualization of examples from Corrupted MNIST dataset (Mu & Gilmer, 2019) and the corresponding reconstructed images for each model are shown in Figure 7 and Figure 8.

Figure 7: Examples of Corrupted MNIST and the reconstructed image for each model. A red box represent that this input is flagged as an adversarial example while a green box represent this input has been misclassified and not been detected.

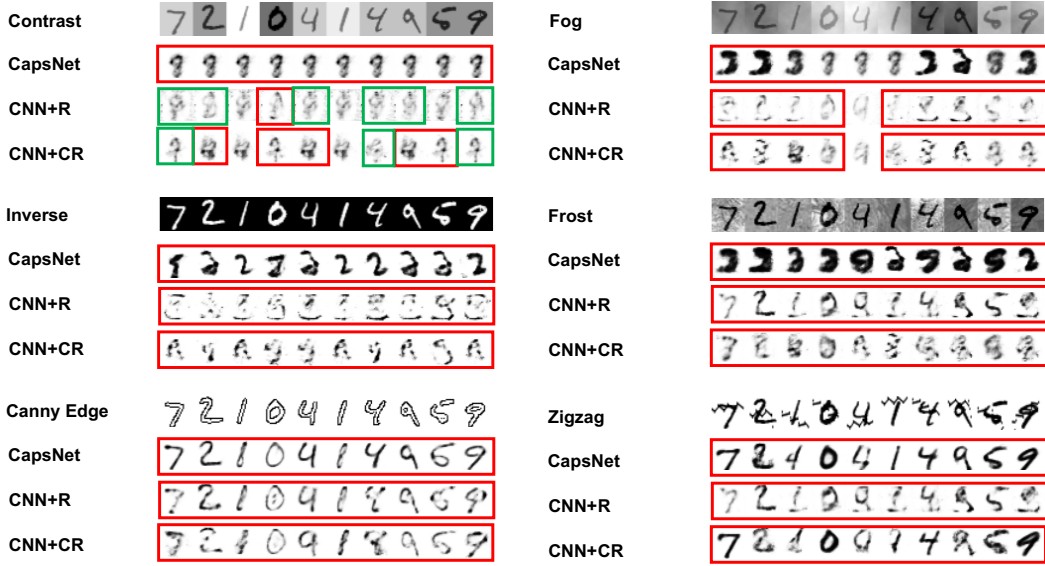

Figure 8: Examples of Corrupted MNIST and the reconstructed image for each model. A red box represents that this input is flagged as an adversarial example while a green box represents that this input has been misclassified and not been detected.

## F RECONSTRUCTIVE ATTACKS

The results of Reconstructive FGSM, BIM and PGD on the three datasets are reported in Table 6.

Table 6: Success rate and the **worst case** undetected rate of white-box targeted and untargeted reconstructive attacks. Below $S_t/R_t$ is shown for targeted attacks and $S_u/R_u$ is presented for untargeted attacks.

| Networks | Targeted (%) | | | Untargeted (%) | | |
|---|---|---|---|---|---|---|
| | R-FGSM | R-BIM | R-PGD | R-FGSM | R-BIM | R-PGD |
| **MNIST Dataset** | | | | | | |
| CapsNet | 1.8/0.3 | 51.0/33.8 | 50.7/33.7 | 6.1/1.0 | 84.5/35.1 | 88.1/37.9 |
| CNN+CR | 7.6/0.5 | 98.0/68.1 | 98.6/68.1 | 41.7/3.2 | 96.5/86.8 | 99.4/87.7 |
| CNN+R | 16.9/3.3 | 86.3/65.9 | 95.5/71.2 | 25.9/8.1 | 82.9/67.8 | 95.1/70.5 |
| **FASHION MNIST Dataset** | | | | | | |
| CapsNet | 6.5/5.8 | 53.3/28.4 | 53.7/29.8 | 33.3/29.9 | 85.3/75.9 | 84.9/75.5 |
| CNN+CR | 17.7/14.0 | 80.3/72.4 | 78.1/72.0 | 68.0/57.3 | 89.8/84.4 | 91.5/86.0 |
| CNN+R | 19.4/17.6 | 95.2/88.8 | 94.6/88.4 | 58.6/53.5 | 98.8/90.1 | 98.9/90.0 |
| **SVHN Dataset** | | | | | | |
| CapsNet | 21.6/21.2 | 81.1/78.3 | 82.0/79.2 | 71.6/68.3 | 98.9/97.5 | 98.9/97.5 |
| CNN+CR | 24.2/22.6 | 98.5/97.6 | 99.0/97.9 | 86.0/82.3 | 99.9/99.5 | 99.9/99.5 |
| CNN+R | 26.6/25.8 | 99.6/99.4 | 99.5/99.3 | 87.1/84.5 | 100.0/99.9 | 100.0/99.9 |

### F.1 HYPERPARAMETER $\beta$

Figure 9 shows the plot of success rate and undetected rate versus the hyperparameter $\beta$ which balances the importance between attacking the classifier and fooling the detection mechanism in the targeted reconstructive PGD attacks on the MNIST dataset.

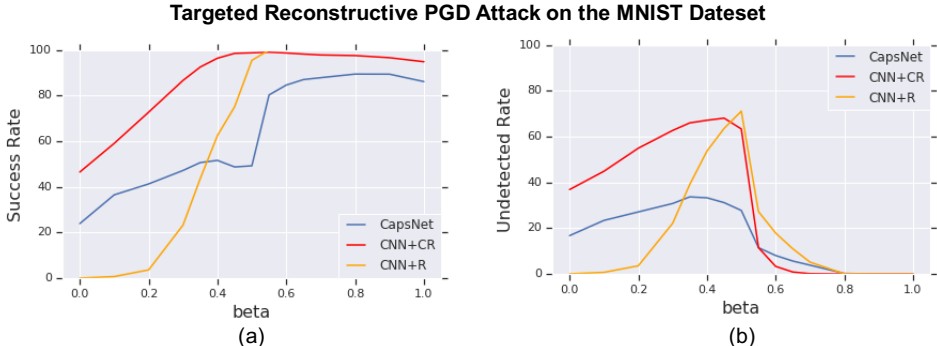

Figure 9: An example shows the plot of the success rate in (a) and undetected rate in (b) of targeted reconstructive PGD attack vesus the hyperparameter beta $\beta$ for each model on the MNIST test set. We set the max $\ell_\infty$ norm $\epsilon = 0.3$ to create the attacks.

## G BLACK BOX ATTACKS

Table 7: Success rate and undetected rate of **black-box** targeted and untargeted attacks. In the table, $S_t/R_t$ is shown for targeted attacks and $S_u/R_u$ is presented for untargeted attacks. All the numbers are shown in %.

| MNIST Dataset | | | | | | | |
|---|---|---|---|---|---|---|---|
| Targeted | CapsNet | CNN-CR | CNN-R | Untargeted | CapsNet | CNN-CR | CNN-R |
| PGD | 1.5/0.0 | 7.8/0.0 | 7.4/0.0 | PGD | 8.5/0.0 | 32.6/0.0 | 27.6/0.0 |
| R-PGD | 4.2/1.0 | 18.3/11.0 | 11.3/4.8 | R-PGD | 10.4/2.4 | 42.7/24.9 | 25.2/8.9 |

## H VISUALIZATION OF ADVERSARIAL EXAMPLES AND RECONSTRUCTIONS

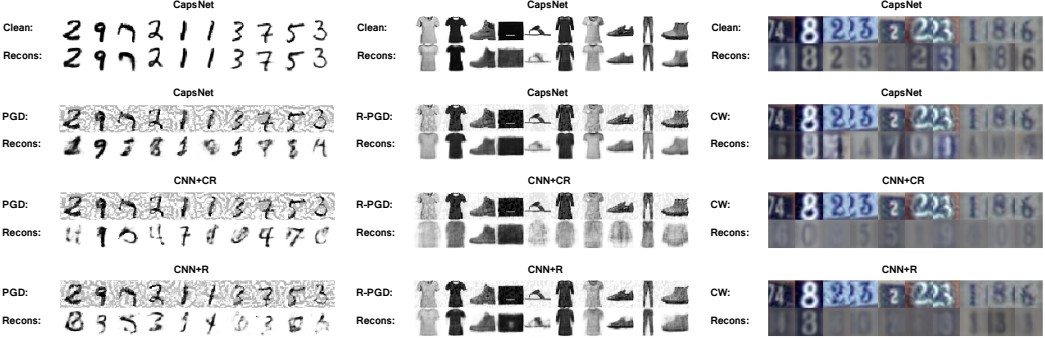

Figure 10: The source clean image is presented in the first row with its reconstruction in the second row. For each model, the top row are the targeted adversarial examples and the bottom are the corresponding reconstruction image when the input are the PGD on the MNIST (left), R-PGD on the Fashion-MNIST (middle), CW on the SVHN (right).

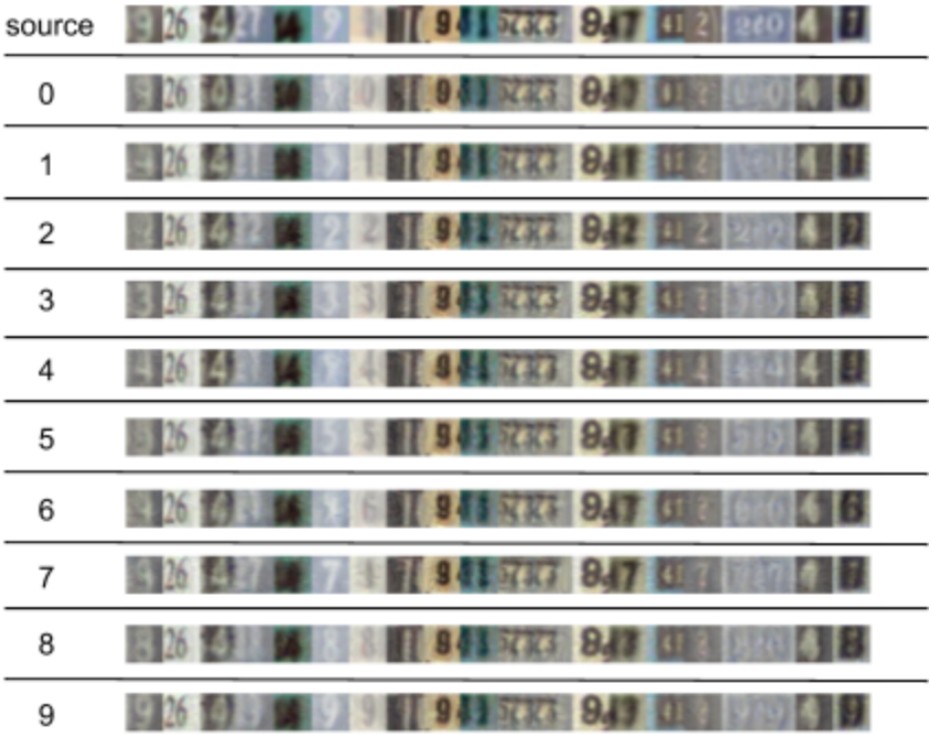

Figure 11: These are randomly sampled (not cherry picked) inputs (top row) and the result of adversarially perturbing them with targeted R-PGD against the CapsNet model (other rows). Many of these attacks are not successful. Note the visual similarity between many of the attacks and the target class.

