# OpenReview forum: "Detecting and Diagnosing Adversarial Images with Class-Conditional Capsule Reconstructions"
_ICLR.cc/2020/Conference — Accept (Poster)_

### Official Review · AnonReviewer3 · 2019-10-23
**Official Blind Review #3**

**Rating:** 6

**Review:**

This paper proposed a new defense method for capsule networks. For both white-box and black-box settings, the proposed CapNets has shown superior performance than two variants of CNNs. The visualizations of adversarial examples generated by the CapNets are more aligned with the human perception which is very insightful. On the corrupted MNIST dataset, the results show the proposed defense method can also be used well as an out-of-distribution detector. Overall the paper is clearly written and easy to follow.



Here I have some concerns:

The major one is the limited comparisons. For the defense of CNNs, the author only implemented two types of the strategy, which is derived from the characteristics of the capsule network. However, there are also other a lot of defense methods for CNNs which are designed according to the characteristics of themselves. I think the author should also compare the defense performance with those methods to hold the strong claims that CpasNets always perform better than convolutional networks. Also, the experiments performed on Cifar-10 is very limited.



The adversarial examples generated by the CapsNets shown in Figure 3 and Figure 11 are indeed changing the number shape which is aligned with human perception. However, some studies for CNNs has also found similar results on MNIST(Towards Deep Learning Models Resistant to Adversarial Attacks) and CIFAR-10(RANDOM MASK: Towards Robust Convolutional Neural Networks). The author should provide more visualizations on other datasets such as CIFAR-10 to support the contribution that the features captured by CapsNets are more aligned with human perception than CNNs.

=========================================================
After Rebuttal:

I thank the author for the response.

I still think the argument that CpasNets always perform better than convolutional networks is an overstatement since you only performed a few defense methods. A milder one is more suitable.

Also, the high variance of the Capsule Network in Figure 10 can tell something but it is not enough. Could you find similar visible semantic changes in CIFAR-10 dataset as Figure 5?  If yes, you should also list some results. The mentioned CNN works could find similar phenomena on both MNIST and CIFAR-10 dataset. I do not see the strong evidence for the argument that features captured by CapsNets are more aligned with human perception than CNNs.

To sum up, I think some arguments are overstatements. But this is a good work to analyze the robustness of Capsule Net, I would like to rate 6.

**Experience Assessment:**

I have published one or two papers in this area.

**Review Assessment: Checking Correctness Of Derivations And Theory:**

N/A

**Review Assessment: Checking Correctness Of Experiments:**

I carefully checked the experiments.

**Review Assessment: Thoroughness In Paper Reading:**

I read the paper thoroughly.

---

> ### Author Response · Authors · 2019-11-14
> **Response to Reviewer 3**
>
> Thanks very much for your suggestions and we believe that most of your concerns are about the performance of our model on the CIFAR-10 dataset. Therefore, we take your advice to add more experiments on the CIFAR-10 dataset in Section 5.4 in the paper to validate the effectiveness of our methods.
>
> First, we compare our models to state-of-the-art CNN-based models against defense-aware attacks (the defense-aware attacks are the most important attack in evaluating the performance of detection methods) and the results are shown in Table 4. We can see that our deflecting model based on CapsNets has a much better detection performance against defense-aware attacks. We should emphasize that most of the existing detection models based on CNNs are further fully attacked by defense-aware attacks. Our detection method based on CapsNets can still have a good detection performance against defense-aware attacks.
>
> In addition, we also plot the white-box and black-box Undetected Rate vs False Positive Rate of the defense-aware R-PGD attack by sweeping the threshold of the l2 reconstruction error from [0, 30] ( Please see Figure 3 (left)). We can see that the black-box R-PGD has a much smaller undetected rate compared to the white-box R-PGD, which implies that the CapsNet greatly reduces the attack transferability.
>
> Furthermore, we include a comparison among our CapsNet with class-conditional reconstruction, “DeepCaps” proposed in [1] and “CapsNet All” using all the capsule information rather than the winning-capsule information on the  CIFAR-10 dataset. The classification performance of the three models on the clean test set is very similar, around 92.5%. The undetected rate of the defense-aware R-PGD versus the False Positive Rate is displayed in Figure 3 (right). We can see that
> 1) our CapsNet with class-conditional information has the smallest Undetected Rate among the three models.
> 2) The CapsNet using all the capsule information has the worst detection performance. This is also consistent with CNN-based models on the other three datasets. As shown in Figure 2, CNN with class-conditional information CNN+CR always has a better detection performance compared to CNN without class-conditional reconstruction CNN+R.
>
> Finally, we also add the visualization of the adversarial success rates for each source/target pair for adversarial attacks on CIFAR-10 with a l_infinity bound 8/255 in Figure 10 in the Appendix. Similarly, we can see that there is a higher variance for the CapsNet model than CNN models. Therefore, we claim that these visualizations on the Fashion-MNIST, SVHN and CIFAR-10 datasets suggest that the features captured by CapsNets are more aligned to human perception compared to CNNs.
>
> [1] Rajasegaran, Jathushan, et al. "DeepCaps: Going Deeper with Capsule Networks." Proceedings of the IEEE Conference on Computer Vision and Pattern Recognition. 2019.

---

### Official Review · AnonReviewer2 · 2019-10-24
**Official Blind Review #2**

**Rating:** 8

**Review:**

This work proposes to detect adversarial examples or otherwise corrupted images with the reconstruction network, which is used to regularise CapsNets. To further confirm the effectiveness of their detection method, they propose the Reconstructive Attack, which seeks both to cause a misclassification and a low reconstruction error. The comprehensive experiments are conducted.

Although the idea is not very novel, the paper makes enough contributions to get accepted. Especially, Section 6 diagnoses the adversarial examples for CapsNets and shows the relationship between the success of the attack and the visual similarity between the source and target class.

We have the following question for authors about this work:

1. Baseline models: This work creates the baseline model CNN+CR, by dividing the penultimate hidden layer of a CNN into groups corresponding to each class. The sum of each neuron group serves as the logit for that particular class. Why is the sum operation used to create the logit? The sum operation is pretty rare the existing CNN architectures. A linear combination may be a better choice?

2. Detection Threshold: this paper empirically chooses the 95th percentile of validation distances as the threshold to detect adversary samples. Why not report the Area Under Curve (AOC) score? It is a more comprehensive evaluation metric for such a problem.

3. Class-conditional information: This paper follows the architecture in Sabour et al. (2017) where the reconstruction is class-conditional. The work DeepCaps [1] shows that the reconstruction without class-conditional information leads to the better disentanglement of instantiation parameters. Is the class-conditional information necessary to achieve the robustness to adversary attacks?
[1] Rajasegaran, Jathushan, et al. "DeepCaps: Going Deeper with Capsule Networks." Proceedings of the IEEE Conference on Computer Vision and Pattern Recognition. 2019.

**Experience Assessment:**

I have read many papers in this area.

**Review Assessment: Checking Correctness Of Derivations And Theory:**

I assessed the sensibility of the derivations and theory.

**Review Assessment: Checking Correctness Of Experiments:**

I assessed the sensibility of the experiments.

**Review Assessment: Thoroughness In Paper Reading:**

I read the paper thoroughly.

---

> ### Author Response · Authors · 2019-11-14
> **Response to Reviewer 2**
>
> Thanks very much for your very positive feedback and support for our work. The answers to your questions are listed below:
>
> 1. We use the simple sum operation on each neuron group to serve as the logit because 1) the sum operation has already enabled the CNN classifier to have a comparable prediction performance to CapsNets.  2) We control the number of parameters in CapsNets and CNNs to be exactly the same. Therefore, the simple sum operation will not incorporate extra parameters compared to CapsNets. Since our target is to compare the performance between CapsNet and CNN-based networks against adversarial examples, we make sure the classification performance and the number of parameters for different network architectures are similar for a fair comparison. 3) In addition, we hypothesize the linear relation of each neuron in each neuron group could be learned by the previous layer to some extent. Therefore, we do not think a linear combination would incorporate a big difference in the results reported in our work.
>
> 2. Thanks very much for your suggestion to draw the AUC curve. We added a plot of the Undetected Rate of the defense-aware R-PGD attack vs the False Positive Rate curve in Figure 2 by changing the threshold of the l2 reconstruction error to measure the performance of different detection methods. We should note that the standard AUC curve plots the True Positive Rate vs False Positive Rate. Since the success rate for the adversarial examples is close to 100%, the AUC curve is enough to measure detection performance. However, in our CapsNets-based model, the success rate (bounded by a small l_infinity norm) is far from 100%. A weak attack that has a small success rate may have a substantially smaller True Positive Rate compared to a stronger attack (The smaller the True Positive Rate is, the stronger the attack is). In this case, the AUC curve is not sufficient to compare different attacks and detection methods. Therefore, we added a plot of the Undetected Rate of the defense-aware R-PGD attack vs the False Positive Rate curve in Figure 2 by changing the threshold of the l2 reconstruction error from [0, 20] for MNIST, [0, 40]  for Fashion-MNIST and [0, 300] for SVHN. Similarly, we can see that CapsNets perform much better than CNN-based networks. That is: at the same False Positive Rate, the defense aware R-PGD attack against CapsNets always has the smallest Undetected Rate (smaller is better for the detection method).
>
>
> 3. Thanks very much for pointing out the class-conditional information. We add a section “Class-conditional Information” in Section 5.4, where we compare our CapsNet with “DeepCaps”[1] and “CapsNet All”. The classification performance of the three models on the clean test set is very similar, around 92.5%. The undetected rate of the defense-aware R-PGD versus the False Positive Rate is displayed in Figure 3 (right). We can see that
> 1) our CapsNet with class-conditional information has the smallest Undetected Rate among the three models.
> 2) The CapsNet using all the capsule information has the worst detection performance. This is also consistent with CNN-based models on the other three datasets. As shown in Figure 2, CNN with class-conditional information CNN+CR always has a better detection performance compared to CNN without class-conditional reconstruction CNN+R.
> 3)  DeepCaps has a little worse detection performance compared to our CapsNet.
> Therefore, we conclude that the class-conditional information is necessary to achieve robustness to adversarial attacks. We include these analyses in Section 5.4.
>
> [1] Rajasegaran, Jathushan, et al. "DeepCaps: Going Deeper with Capsule Networks." Proceedings of the IEEE Conference on Computer Vision and Pattern Recognition. 2019.

---

### Official Review · AnonReviewer1 · 2019-10-28
**Official Blind Review #1**

**Rating:** 6

**Review:**

This paper studies the problem of detecting and generating adversarial images using class-conditional capsule networks. Specifically, this paper first introduced a novel method that detects adversarial examples by class-conditional image reconstruction. Motivated by this defense method, this paper further proposed a novel reconstructive attack that minimizes both classification and reconstruction loss. Experimental evaluations are conducted on MNIST, FashionMNIST, SVHN, and CIFAR-10 dataset. Results demonstrate the effectiveness of the proposed defense and the novel reconstructive attack method.

Overall, this paper is well-motivated and presentation is clear. It proposed a smart way of using the class-conditional generative model to improve the adversarial robustness. Please address the following questions.

(1) Reviewer’s major concern is that this method is not very scalable to large-scale real-world datasets such as ImageNet and SUN database. First, training a class-conditional generative model on MNIST is relatively easy compared to ImageNet. The generative model could potentially create image artifacts on higher resolution images.

-- SUN Database: Large-scale Scene Recognition from Abbey to Zoo. Xiao et al. In CVPR 2010.

Second, the proposed proxy based on l_2 image distance might not be effective at all for higher-resolution images.

-- A note on the evaluation of generative models, Theis et al. In ICLR 2016.
-- The Unreasonable Effectiveness of Deep Features as a Perceptual Metric, Zhang et al. In CVPR 2018.

(2) Reviewer is not fully convinced by the argument that features learned by CapsNets are superior to features learned by CNN baselines. To draw such conclusion, it is necessary to conduct systematic experiments with different CapsNets and CNNs architectures (e.g., number of layers) and other hyper-parameters related to the adversarial optimization.

(3) It looks like the proposed method is not specific to generative models use class labels as condition. reviewer is curious whether the method generalizes to other conditions (e.g., image-to-image translation) as well.

-- Learning Structured Output Representation using Deep Conditional Generative Models, Sohn et al. In NIPS 2015.
-- Image-to-Image Translation with Conditional Adversarial Nets, Isola et al. In CVPR 2017.
-- Unpaired Image-to-Image Translation using Cycle-Consistent Adversarial Networks, Zhu et al. In ICCV 2017.
-- Semantic Image Synthesis with Spatially-Adaptive Normalization, Park et al. In CVPR 2019.

(4) Can you possibly comment on the attack transferability compared to other existing attacks evaluated in this paper?

(5) Detection threshold: Can you possibly draw the AOC curve or report the Area Under Curve (AOC) score?


**Experience Assessment:**

I have published in this field for several years.

**Review Assessment: Checking Correctness Of Derivations And Theory:**

I assessed the sensibility of the derivations and theory.

**Review Assessment: Checking Correctness Of Experiments:**

I carefully checked the experiments.

**Review Assessment: Thoroughness In Paper Reading:**

I read the paper thoroughly.

---

> ### Author Response · Authors · 2019-11-14
> **Response to Reviewer 1**
>
> We really appreciate the reviewer’s positive feedback and helpful suggestions. We answer your questions as follows:
>
> 1. We agree with the reviewer’s opinion that our current method has not been scaled to a large-scale real-word dataset such as ImageNet. However, we should note that Capsule Networks have never been scaled to a large dataset like ImageNet. Our work successfully scaled CapsNets to the CIFAR-10 dataset with state-of-art classification accuracy of 92.2% on the clean test set and we found that our methods generalize very well from MNIST, SVHN to CIFAR-10, which suggests it is very promising to make our detection methods generalize to even larger datasets once CapsNets are scaled to ImageNet. We also appreciate the reviewer’s suggestion of an even better similarity metric rather than l_2 distance. We believe these are promising avenues for future work and we add them to Section 7 in the paper.
>
> 2. To address the concern that “it is not fully convincing that features learned by CapsNets are superior to features learned by CNN baselines”:
> (1) Our intention is not to claim that the learned features are superior. Instead, what we conclude is that the experimental results on MNIST, Fashion-MNIST, SVHN and CIFAR10 “suggest” that features learned by CapsNet may be more robust to adversarial attack and may be more aligned to human perception.
> (2) Second, the network architecture (eg. number of layers) and the hyper-parameters related to adversarial optimization are different for each dataset.
> (3) Lastly, apart from the visualization on the Fashion-MNIST and SVHN datasets displayed in Section 6 in the paper, we also add the visualization of  the adversarial success rates for each source/target pair for adversarial attacks on CIFAR10 with an l_infinity bound of 8/255 in Figure 10 in Appendix. Similarly, we can see that there is a higher variance for the CapsNet model than CNN models.
>
> 3.  We totally agree with the reviewer that the proposed method is not specific to generative models using class labels for conditioning. We believe it is very promising for our method to generalize to other conditions  (e.g., image-to-image translation) as well. We add this as a promising and interesting future work in Section 7.
>
> 4.  For our proposed defense-aware R-PGD attack, we did not observe a significant difference compared to other existing attacks. However, we observe that the attack transferability against CapsNets will be greatly reduced compared to CNN based networks.
>
> 5.  We take the reviewer’s advice to draw the AUC curve to measure the performance of different detection methods in Figure 2. We should note that the standard AUC curve plots the True Positive Rate vs False Positive Rate. Since the success rate for standard adversarial examples is close to 100%, the AUC curve is enough to measure detection performance. However, in our CapsNets-based model, the success rate (bounded by a small l_infinity norm) is far lower than 100%. A weak attack that has a small success rate may have a substantially smaller True Positive Rate compared to a stronger attack (The smaller the True Positive Rate is, the stronger the attack is). In this case, the AUC curve is not sufficient to compare different attacks and detection methods. Therefore, we have added a plot of the Undetected Rate of the defense-aware R-PGD attack vs the False Positive Rate curve in Figure 2 by changing the threshold of the l2 reconstruction error from [0, 20] for MNIST, [0, 40]  for Fashion-MNIST and [0, 300] for SVHN. Similarly, we can see that CapsNets perform much better than CNN-based networks. That is: at the same False Positive Rate, the defense aware R-PGD attack against CapsNets always has the smallest Undetected Rate (smaller is better for the detection method).

---

### Author Response · Authors · 2019-11-14
**General Response**

We thank all the reviewers for their constructive and positive feedback and take the reviewer’s suggestions including more experiments and adding the corresponding analysis in the paper. They are mainly:

1) We plot the undetected rate versus the False Positive Rate by changing the threshold on four datasets, see Figure 2.
2) We include the comparison with other state-of-the-art CNN based models on CIFAR-10 dataset, see Table 4.
3) We add the visualization of the success rate between source/target pairs on the CIFAR-10 dataset, see Figure 10 in the Appendix.
4) We compare our CapsNet with two other CapsNet variants on CIFAR-10 dataset: one is a CapsNet without masking mechanism and another one is a CapsNet with class-independent reconstruction[1], see Figure 3 (right).

Please feel free to let us know if you have other questions and thanks again for your helpful suggestions.

[1] Rajasegaran, Jathushan, et al. "DeepCaps: Going Deeper with Capsule Networks." Proceedings of the IEEE Conference on Computer Vision and Pattern Recognition. 2019.

---

### Decision · Program_Chairs · 2019-12-19

**Decision:**

Accept (Poster)

**Comment:**

This paper presents a mechanism for capsule networks to defend against adversarial examples, and a new attack, the reconstruction attack. The differing success of this attacks on capsnets and convnets is used to argue that capsnets find features that are more similar to what humans use.

Reviewers generally like the paper, but took instance with the strength of the claim (about the usefulness of the examples) and argued that the paper might not be as novel as it claims.

Still, this seems like a valuable contribution that should be published.